# Blood donations and donors' profile in Lithuania: Trends for coming back after the COVID-19 outbreak

**Vytenis Kalibatas** [1]*, **Lina Kalibatienė** [2], **Dulat Imashpayev** [3]

1 Department of Health Management, Lithuanian University of Health Sciences, Kaunas, Lithuania,
2 Department of Anaesthesiogy, Lithuanian University of Health Sciences, Kaunas, Lithuania, 3 Scientific and Production Center of Transfusiology, Astana, Kazakhstan

☯ These authors contributed equally to this work.
* vytenis.kalibatas@lsmu.lt

**Data Availability Statement:** All relevant data are within the manuscript.

**Funding:** The author(s) received no specific funding for this work.

## Abstract

The coronavirus disease (COVID-19) pandemic has significantly affected blood donors worldwide. It is important for the blood service to return to its pre-pandemic level as soon as possible and to perform its functions fully. This study compared the donation and demographic profiles of blood and its component donors one year before and during three pandemic years in Lithuania. All blood and blood component donations (n = 413,358) and demographic characteristics of all donors from April 1, 2019, to March 31, 2023, were analyzed. All data were obtained from annual publications, and statistics were obtained from the Blood Donor Register. Data were analyzed using descriptive statistics. Following a 9.41 percent decrease in the first year of the pandemic, the quantity of blood and blood component donations increased by 3.49 percent in the third year compared to the pre-pandemic year. Throughout the three years of the pandemic, a statistically significant decrease in the proportion of first-time blood and blood component donations was observed. Both the number and proportion of donations by donors under 25 years old decreased during the pandemic. The proportion of pre-donation deferrals for all attempts to donate significantly decreased during the pandemic. There was a statistically significant lower prevalence of all positive transfusion-transmitted infectious (TTI) markers among all donations compared to the pre-pandemic year for all three pandemic years. The odds for hepatitis B virus, hepatitis C virus, and all TTI markers during the second and third pandemic years were significantly lower than those in the pre-pandemic year. In conclusion, most dimensions of blood and its component donations and donor characteristics have returned to pre-pandemic levels or show positive trends. However, the major concern is the remaining decrease in donations from first-time and donors under 25 years old.

## Introduction

The first cases of novel coronavirus were first detected in China in December 2019, with the virus spreading rapidly to other countries worldwide. This led the WHO to declare a Public

**Competing interests:** The authors have declared that no competing interests exist.

Health Emergency of International Concern (PHEIC) on January 30, 2020, and to characterize the outbreak as a pandemic on March 11, 2020. On May 5, 2023, the WHO Emergency Committee on COVID-19 recommended recognizing that, given that the disease was well-established and ongoing, it no longer fit the definition of a PHEIC [1]. Many aspects of human life have been disrupted by the COVID-19 pandemic [2], which had massive consequences on societies and health systems across countries [3].

Blood transfusions are a vital component of every country's health service [4]. Ensuring a sufficient and safe supply of blood and blood products is vital for the effective functioning of healthcare systems and patients' overall well-being. However, ensuring a sufficient supply of safe blood may prove challenging because disasters or other emergencies may damage the available civil and healthcare infrastructure, disrupting mobility, transportation, and service provision. Moreover, some population members may not be able to donate blood because of fear or illness [5]. The COVID-19 pandemic containment measures imposed to mitigate the spread of COVID-19, including lockdown, social distancing guidelines, and travel restrictions, have disrupted regular blood donation processes. This has caused a significant decrease in the number of blood donations worldwide and has become a serious issue [6]. A few scientific studies have revealed that the profile of donors changed during the pandemic period: the COVID-19 pandemic impacted the type (first time/repeat or regular) of blood donations and demographic profile (age and sex) of blood donors [7, 8]. Other studies have analyzed blood safety in terms of the prevalence of infectious disease markers [9, 10] and donor deferrals to donate blood and its components during the COVID-19 pandemic [11].

Although considerable publications are focusing on the various consequences of the COVID-19 pandemic on blood transfusion services, there are no publications on the impact of the COVID-19 pandemic on blood donations and donor profiles over an extended period, revealing important trends after drastic changes that occurred during the initial stages of the pandemic.

This study aimed to assess the impact of the COVID-19 pandemic on the number, type, and safety of blood and its components' donations and the demographic profile of blood donors, comparing the years before and during the pandemic and examining trends over three pandemic years.

## Materials and methods

### Data collection

All blood and blood component donations and the demographic characteristics of all donors in Lithuania from April 1, 2019, to March 31, 2023, were analyzed. All data were obtained from statistical reports of the National Blood Donors´ register [12]. During this study, the following dimensions were analyzed: the total number of blood and its components donated; number of donations according to donation type (first-time/regular or repeat); donors' demographic characteristics (age and sex); procedure type (whole blood, red blood cell apheresis, platelet apheresis, plasmapheresis); the prevalence of HIV 1 and 2, HBV, HCV, and syphilis markers per 100 donations; the causes of donation restrictions; and temporary deferrals.

### Study period

The study period was 4 years. Like most countries, Lithuania declared quarantine in mid-March 2021 and restricted regular human and institutional activities [13]. The quarantine was revoked from June 1, 2021; however, the state level of emergency in the country ceased to be in effect from May 1, 2022 [14]. Thus, we determined that the one-year study period would start on April 1 and end on March 31. The period from April 1, 2019, to March 31, 2020, was

considered the pre-pandemic year; the period from April 1, 2020, to March 31, 2021, was considered the first year of the pandemic; the period from April 1, 2021, to March 31, 2022, was considered the second year of the pandemic; and the period from April 1, 2022, to March 31, 2023, was considered the third year of the pandemic.

## Donor screening and deferral

A first-time donor is defined as an individual who has not previously donated blood or its components. A repeat donor is an individual who has donated blood or its components once within a year, with the penultimate donation occurring two years or more prior. A regular donor is an individual who regularly donates blood or its components (at least two or more times within a two-year period).

Blood and blood components were tested for serological anti-HCV, HBV surface antigen (HBsAg), Ag/anti-HIV 1 and 2, and syphilis antibodies. If the results of serological screening were positive or uncertain, the individual sample test was repeated. If any of the repeated tests yielded positive or uncertain results, the subsequent confirmatory tests employed were as follows: the HBsAg neutralization test for HBV, the anti-HCV immunoblot test for HCV, Western blotting and/or HIV 1–p24 antigen and/or HIV 2 ELISA for HIV 1 and 2, and the *Treponema Pallidum* hemagglutination (TPHA) test for syphilis. Seronegative donors were tested for HCV, HBV, and HIV-1 infections using a nucleic acid test (NAT). If the results of individual samples were reactive for HIV RNA, HCV RNA, or HBV DNA, a confirmatory quantitative test was performed [15]. Donations with confirmed infectious disease markers (serological or NAT) were considered positive.

The criteria for permanent donation restriction and temporary deferrals of blood and its components were set up by the Ministry of Health of the Republic of Lithuania [16]. Donors of blood or its components may be permanently restricted from donating for one or more of the following reasons: circulatory system disease, disease of the central nervous system, increased risk of bleeding, repeated fainting cases or former seizures, diabetes, malignant diseases, use of intravenous drugs and/or oral/intramuscular anabolic steroids or hormonal substances, and some infectious and other diseases. Temporary deferrals involve the following reasons: the current physiological and health status of the donor (age under 18 years, weight, inappropriate hemoglobin level, blood pressure or heart rate, fever, rheumatism, some infectious and other diseases), recent surgical operations or procedures, tattooing and ring piercing, vaccination, pregnancy, and the use of some medicines. Deferral periods vary for various reasons.

## Statistical analysis

The z-test was used to compare two proportions. Prevalence was calculated as the ratio of the number of confirmed positive results for transfusion-transmitted infectious (TTI) disease markers (serological syphilis, anti-HCV, HBsAg, anti-HIV 1 and 2, and/or HCV, HBV, and HIV-1 NAT) per 100 donations. Odds ratios (OR) and 95% confidence intervals (CIs) were determined. Statistical significance was set at P <0.05. Data analysis was conducted using IBM SPSS software (version 29,0).

## Ethical considerations

As per local legislation, this survey is not the subject of bioethical regulation because generalized, fully anonymized, and publicly available data (as opposed to collecting personal data directly from research participants, medical health records, or archived samples) have been used. The Kaunas Regional Biomedical Research Ethics Committee reviewed the methodology of this survey and deemed the investigation an evaluation of service, not requiring review by

an ethics committee, as the direct object of this study was not a specific person or person´s health. The analysis of de-identified, fully anonymized, generalized, and publicly available data did not constitute human subject research and did not require the participants' consent.

## Results

Fig 1 shows the number of blood and blood components' donations in Lithuania between April 2019 and March 2023.

A cumulative count of 105,321 blood and component donations was recorded during the pre-pandemic period in Lithuania. During the first year of the COVID-19 pandemic, blood and blood component donations decreased by 9.41 percent, and during the second year, they decreased by 1.60 percent compared to pre-pandemic year. In the third year of the pandemic, the number of blood and blood component donations increased by 3.49 percent compared with the pre-pandemic year (Table 1).

In the pre-pandemic year, a total of 361 blood and its component donations were remunerated (constituting 0.34 percent of all donations). In the second year, this number dropped significantly to four donations (0.004 percent of all donations) in Lithuania. Notably, no remunerated donations were recorded during the second and third years of the pandemic.

Throughout the three years of the COVID-19 pandemic, a statistically significant decrease in the proportion of first-time blood and blood component donations was observed. The proportion of first-time donations was significantly lower during the first (z = 40.86; P<0.001), second (z = 36.39, P<0.001), and third (z = 35.3, P<0.001) pandemic years compared to the pre-pandemic year and did not reach the pre-pandemic level. The proportion of first-time donations during the first pandemic year was statistically significantly lower compared with the second (z = 5.49, P<0.001) and the third (z = 7.16, P<0.001) pandemic years. Despite this positive trend, neither the number nor the proportion of first-time donations reached the pre-

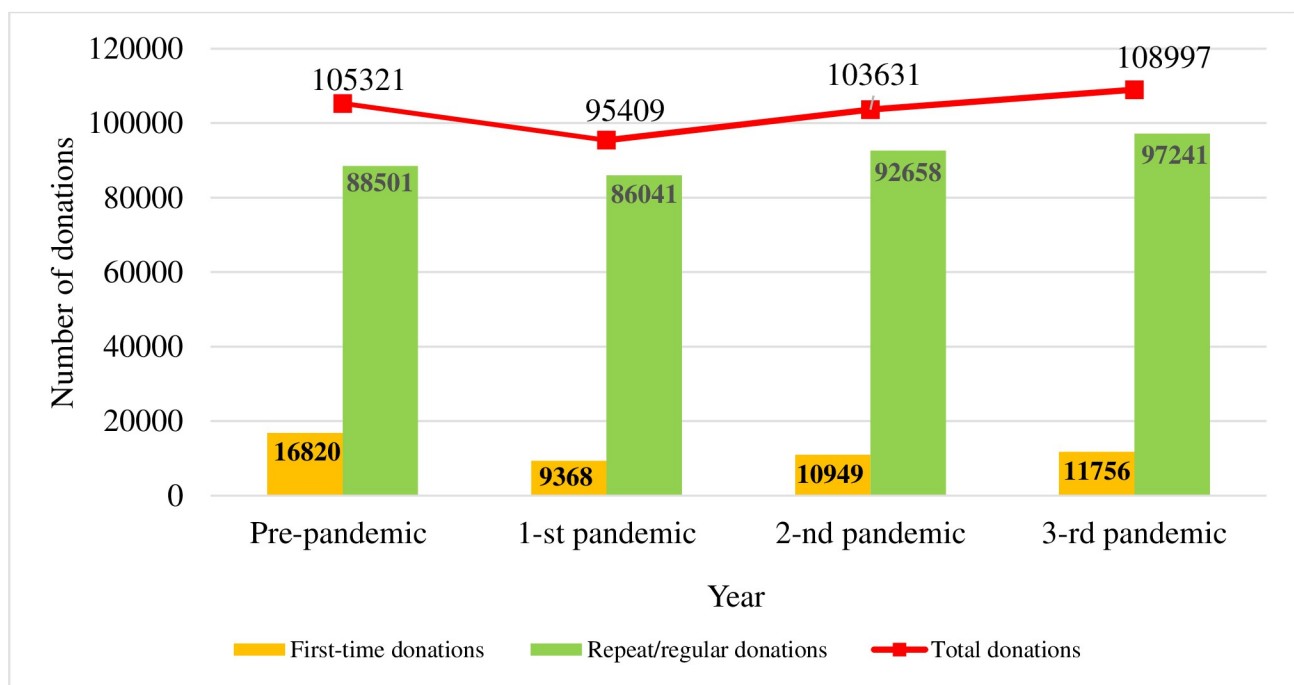

**Fig 1. Number of blood and blood components' donations in Lithuania between April 2019 and March 2023.**

**Table 1. Number of blood and its components' donations according to specific dimensions in Lithuania between April 2019 and March 2023.**

| Dimensions of donations | Years | | | | % change (+/-), comparing Pre-pandemic year with | | |
|---|---|---|---|---|---|---|---|
| | Pre-pandemic No (%*) | 1st pandemic No (%*) | 2nd pandemic No (%*) | 3rd pandemic No (%*) | 1st pandemic | 2nd pandemic | 3rd pandemic |
| Total blood and components donations | 105321 | 95409 | 103631 | 108997 | -9.41 | -1.60 | +3.49 |
| *Type of remuneration* | | | | | | | |
| Voluntary, non-remunerated | 104870 (99.66)[a] | 95405 (99.996)[b] | 103631 (100.0) | 108997 (100.0) | -9.02 | -1.18 | +3.93 |
| Remunerated | 361 (0.34)[a] | 4 (0.004)[b] | 0 (0.0) | 0 (0.0) | -98.89 | N/A | N/A |
| *Type of blood donor* | | | | | | | |
| First-time | 16820 (15.97)[a] | 9368 (9.82)[b] | 10949 (10.57) | 11756 (10.79) | -44.30 | -34.90 | -30.11 |
| Repeat/regular | 88501 (84.03)[a] | 86041 (90.18)[b] | 92658 (89.43) | 97241 (89.21 | -2.78 | +4.70 | +9.86 |
| *Place of donation* | | | | | | | |
| Blood establishment | 34787 (33.03)[a] | 46871 (49.13)[b] | 46692 (45.06)[c] | 46231 (42.41) | +34.74 | +34.22 | +32.89 |
| Mobile session | 70534 (66.97)[a] | 48538 (50.87)[b] | 56939 (54.94)[c] | 62766 (57.59) | -31.18 | -19.27 | -11.01 |
| *Type of donation* | | | | | | | |
| Whole blood | 99664 (94.63)[c] | 88818 (93.09)[b] | 98031 (94.60)[c] | 103508 (94.96) | -10.88 | -1.64 | +3.86 |
| Double RBC apheresis | 2764 (2.62)[a] | 3141 (3.29)[b] | 2535 (2.45)[c] | 2380 (2.18) | +13.64 | -8.29 | -13.90 |
| RBC apheresis | 31 (0.029)[a] | 12 (0.01)[c] | 6 (0.01) | 3 (0,003) | -61.29 | -80.65 | -90.32 |
| Plasmapheresis | 1 (0.001)[a] | 766 (0.80)[b] | 247 (0.24) | 233 (0.217) | +>100 | +>100 | +>100 |
| Platelet apheresis | 2747 (2.61)[d] | 1747 (1.83)[b] | 2643 (2.55)[c] | 2520 (2.31) | -36.40 | -3.79 | -8.26 |
| Plasma and platelet apheresis | 114 (0.11)[a] | 925 (0.97)[b] | 169 (0.16)[c] | 353 (0.32) | +>100 | +48.25 | +>100 |

*From total donations during particular year

[a]–P<0.05 comparing with the 1-st, 2-nd and 3-rd pandemic years

[b]-P<0.05 comparing with the 2-nd and 3-rd pandemic years

[c]-P<0.05 comparing with the 3-rd pandemic year

[d]-P<0.05 comparing with the 1-st and 3-rd pandemic years

pandemic levels. There was also a decrease in blood and blood component donations during mobile blood-driving. The proportion of donated blood during mobile blood drives was significantly lower during the first (z = 73.32, P<0.001), second (z = 56.36, P<0.001), and third pandemic years (z = 44.8, P<0.001) than during the pre-pandemic year. The proportion of donated blood during mobile blood drives during the first-pandemic year was statistically significantly lower compared witgh the second (z = 18.18, P<0.001) and the third (z = 30.39, P<0.001) pandemic years, while the proportion of donations during mobile blood drives during the second pandemic year was statistically significantly lower compared with the third pandemic year (z = 12.27, P<0.001). Despite this positive trend, neither the number nor the proportion of donations in mobile sessions reached pre-pandemic levels.

Compared to the pre-pandemic years, statistically significant differences were observed in the proportion of donations based on the item being donated. The proportion of whole blood donations to the total number of donations was significantly lower than that in the pre-pandemic year (z = 14.34, P<0.001). The proportion of whole blood donations during the first pandemic year was statistically significantly lower compared with the second (z = 13.95, P<0.001) and the third (z = 17.88, P<0.001) pandemic years, while the proportion of whole blood donations during the second pandemic year was statistically significantly lower compared with the third pandemic year (z = 3.38, P<0.001). Moreover, the proportion of platelet apheresis donations was significantly lower during the first year of the pandemic than during the pre-pandemic year (z = 11.8, P<0.001), and lower compared with the second (z = 10.92, P<0.001) and the third (z = 7.59, P<0.001) pandemic years During the third pandemic year,

the proportion of donations based on the item donated was very similar to that in the pre-pandemic year, except for an increase in plasmapheresis and plasma and platelet apheresis donations and a decrease in red blood cell apheresis. The score values of donations proportions' comparisons between each other year (pre-pandemic year compared with the first, second and third pandemic years; the first pandemic year compared with the second and third pandemic years; the second pandemic year compared with the third pandemic year, respectively) according to specific dimensions are provided in S1 Table.

There were statistically significant differences in donor age and sex between the pre-pandemic and pandemic years. Both the number and proportion of donors aged <25 years decreased during the pandemic (Table 2). The proportion of donations of both males and females under 25 years old during the pre-pandemic year were significantly higher than during the pandemic years (z = 27.58, P<0.001; z = 37.58, P<0.001; and z = 52.26, P<0.001, respectively). The proportion of donation of both males and females under 25 years old during the first pandemic year was statistically significantly higher compares with the second (z = 9.20,

**Table 2. Number of donations according to donors' age and sex in Lithuania between April 2019 and March 2023.**

| Blood and its components donors | Years | | | | % change (+/-), comparing Pre-pandemic year with | | |
|---|---|---|---|---|---|---|---|
| | Pre-pandemic No (%*) | 1st pandemic No (%*) | 2nd pandemic No (%*) | 3rd pandemic No (%*) | 1st pandemic | 2nd pandemic | 3rd pandemic |
| *Males* | | | | | | | |
| Under 25 years old | 13047 (12.39)[a] | 9172 (9.61)[c] | 9929 (9.58)[c] | 9036 (8.29) | -29.70 | -23.90 | -30.74 |
| 25–34 years old | 18842 (17.89)[e] | 17826 (18.68)[c] | 19106 (18.44) | 19811 (18.17) | -5.39 | +1.40 | +5.14 |
| 35–44 years old | 16097 (15.28)[a] | 16369 (17.16)[b] | 18195 (17.56)[c] | 20270 (18.60) | +1.69 | +13.03 | +25.92 |
| 45–54 years old | 9500 (9.02)[a] | 9711 (10.18)[b] | 11519 (11.11)[c] | 13219 (12.13) | +2.22 | +21.25 | +39.15 |
| 55–64 years old | 2697 (2.56)[a] | 2744 (2.88)[b] | 3565 (3.44)[c] | 4058 (3.72) | +1.74 | +32.18 | +50.46 |
| 65 and over years old | 43 (0.04)[c] | 35 (0.03)[b] | 60 (0.06) | 66 (0.06) | -18.60 | +39.53 | +53.49 |
| In total | 60226 (57.18)[a] | 55857 (58.54)[b] | 62374 (60.19)[c] | 66460 (60.97) | -7.25 | +3.57 | +10.35 |
| *Females* | | | | | | | |
| Under 25 years old | 10952 (10.40)[a] | 7827 (8.20)[b] | 6927 (6.69)[c] | 6275 (5.76) | -28.53 | -36.75 | -42.70 |
| 25–34 years old | 11445 (10.87)[b] | 10556 (11.07)[b] | 10149 (9.79)[c] | 9949 (9.13) | -7.77 | -11.32 | -13.07 |
| 35–44 years old | 11531 (10.95)[d] | 10759 (11.28) | 11493 (11.09) | 12256 (11.24) | -6.69 | -0.33 | +6.29 |
| 45–54 years old | 8017 (7.61)[a] | 7627 (7.99)[b] | 8955 (8.64)[c] | 9927 (9.11) | -4.86 | +11.70 | +23.82 |
| 55–64 years old | 3126 (2.97)[b] | 2772 (2.91)[b] | 3689 (3.56)[c] | 4066 (3.73) | -11.32 | +18.01 | +30.07 |
| 65 and over years old | 24 (0.02)[b] | 11 (0.01)[b] | 44 (0.04) | 64 (0.06) | -54.17 | +83.33 | +>100 |
| In total | 45095 (42.82)[a] | 39552 (41.46)[b] | 41257 (39.81)[c] | 42537 (39.03) | -12.29 | -8.51 | -5.67 |
| *Both males and females* | | | | | | | |
| Under 25 years old | 23999 (22.79)[a] | 16999 (17.82)[b] | 16856 (16.26)[c] | 15311 (14.05) | -29.17 | -29.76 | -36.20 |
| 25–34 years old | 30287 (28.76)[a] | 28382 (29.75)[b] | 29255 (28.23)[c] | 29760 (27.30) | -6.29 | -3.41 | -1.74 |
| 35–44 years old | 27628 (26.23)[a] | 27128 (28.43)[c] | 29688 (28.65)[c] | 32526 (29.84) | -1.81 | +7.46 | +17.73 |
| 45–54 years old | 17517 (16.63)[a] | 17338 (18.17)[b] | 20474 (19.76)[c] | 23146 (21.24) | -1.02 | +16.88 | +32.13 |
| 55–64 years old | 5823 (5.53)[a] | 5516 (5.78)[b] | 7254 (7.00)[c] | 8124 (7.45) | -5.27 | +24.57 | +39.52 |
| 65 and over years old | 67 (0.06)[b] | 46 (0.05)[b] | 104 (0.10) | 130 (0.12) | -31.34 | +55.22 | +94.03 |

*From total donations during particular year

[a]–P<0.05 comparing with the 1-st, 2-nd and 3-rd pandemic years

[b]-P<0.05 comparing with the 2-nd and 3-rd pandemic years

[c]-P<0.05 comparing with the 3-rd pandemic year

[d]-P<0.05 comparing with the 1-st and 3-rd pandemic years

[e]-P<0.05 comparing with the 1-st and 2-nd pandemic years

P<0.001) and the third (z = 23.31, P<0.001) pandemic years. Moreover, the proportion of both males and females under 25 years old during the second pandemic year was statistically significantly higher compared with the third pandemic year (z = 14.27, P<0.001). The proportions of donations of both males and females 25–34 years old, 35–44 years old, 45–55 years old and 55–64 years old, during the first pandemic year were statistically significantly higher compared with pre-pandemic year. There were found some different trends analyzing the donations of males and females of the same age group. The number of 25–34 years old male donors' donations declined 5.39 percent during the first pandemic year, while the proportion of donations during the first pandemic year was statistically significant higher compared with the pre-pandemic year (z = 4.59, P<0.05). During the second and third pandemic years, the number of 25–34 years old male donors' donations increased, while the proportions decreased. The number and proportions of 25–34 years old female donors' donations declined during all pandemic years, and showed statistically significant decreasing of donation proportions: the proportion of donations during pre-pandemic year was statistically significantly higher compared with the second (z = 8.06, P<0.001) and the third (z = 13.43, P<0.001) pandemic years; the proportion of donations during the first pandemic year was statistically significantly higher compared with the second (z = 9.28, P<0.001) and the third (z = 14.54, P<0.001) pandemic years; the proportion of donations during the second pandemic year was statistically significantly higher compared with the third (z = 5.24, P<0.001) pandemic year. The proportion of male donors showed a tendency to increase during the pandemic years, with statistically significant differences compared with the pre-pandemic year (z = 6.16, P<0.001; z = 13.97, P<0.001; and z = 17.84, P<0.001, respectively) The score values of donations proportions' comparisons between each other year (pre-pandemic year with the first, second and third pandemic years; the first pandemic year with the second and third pandemic years; the second pandemic year with the third pandemic year, respectively) according to donors' age and sex are provided in S2 Table.

During the pre-pandemic year, there were 122509 attempts to donate blood and its components and 17188 pre-donation deferrals (deferral rate, 14.03). During the pandemic, the number of pre-donation deferrals decreased comparing with the pre-pandemic year (Table 3). The proportions of pre-donation deferrals of all attempts to donate significantly decreased during the pandemic years (z = 16.44, P<0.001; z = 8.94, P<0.001; and z = 12.7, P<0.001, respectively) compared with pre-pandemic year.

During the pre-pandemic year, the main reasons for permanent donor deferral were positive TTI disease markers and somatic illnesses. The proportion of positive TTI markers for all deferrals significantly decreased during the third pandemic year compared with the pre-pandemic year (z = 3.63, P<0.001). During the pre-pandemic year, the main reasons for temporary donor deferral were inappropriate hemoglobin levels, inappropriate blood pressure, heart rate and/or rhythm, recent tattoos and/or body piercing, and recent surgery and/or intervention. Despite the number of deferrals due to inappropriate hemoglobin levels during the first pandemic year decreasing by 20.31 percent, the proportion of temporary deferrals significantly increased (z = 7.72, P<0.001) compared with the pre-pandemic year. A similar situation was observed when analyzing the deferrals due to inappropriate blood pressure, heart rate, and/or rhythm. During the first pandemic year, the number of deferrals decreased by 7.79 percent, but the proportion of such deferrals of all temporary deferrals significantly increased (z = 6.35, P<0.001) compared to the pre-pandemic year. The number of deferrals due to recent tattoos and/or body piercing, as well as surgery and/or intervention, decreased by 42.41 and 36.54 percent, respectively, while there was a statistically significant decrease in the proportions of all temporary deferrals (z = 4.35, P<0.001 and z = 2.51, P<0.05, respectively). There were no statistically significant differences in the proportions of temporary deferrals due to inappropriate

**Table 3. Blood and its components donors' deferrals in Lithuania between April 2019 and March 2023.**

| Dimensions | Years | | | | % change (+/-), comparing Pre-pandemic year with | | |
|---|---|---|---|---|---|---|---|
| | Pre-pandemic No (%) | 1st pandemic No (%) | 2nd pandemic No (%) | 3rd pandemic No (%) | 1st pandemic | 2nd pandemic | 3rd pandemic |
| Attempts to donate | 122509 | 108058 | 118890 | 124425 | -11.80 | -2.95 | +1.56 |
| Pre-donation deferrals | 17188 (14.03*)[a] | 12649 (11.71*)[b] | 15269 (12.83*)[c] | 15428 (12.40*) | -26.41 | -11.16 | -10.24 |
| *Permanent deferrals* | | | | | | | |
| Positive TTI markers | 251 (57.44**)[c] | 154 (55.40**)[c] | 174 (51.33**) | 168 (44.68**) | -38.65 | -30.68 | -33.07 |
| Somatic illnesses or conditions | 156 (35.70**) | 87 (31.29**)[c] | 127 (37.46**) | 154 (40.96**) | -44.23 | -18.59 | -1.28 |
| Infectious diseases | 6 (1.37**)[f] | 4 (1.44**)[f] | 0 (0.0**)[c] | 12 (3.19**) | -33.33 | N/A | +100.0 |
| Drug usage | 2 (0.46**) | 1 (0.36**) | 4 (1.18**) | 3 (0.80**) | -50.0 | +100.0 | -25.0 |
| Other reasons | 22 (5.03**)[a] | 32 (11.51**) | 34 (10.03**) | 39 (10.37**) | +45.45 | +54.55 | +77.27 |
| All | 437 | 278 | 339 | 376 | -36.38 | -22.43 | -13.96 |
| *Temporary deferrals* | | | | | | | |
| Inappropriate level of hemoglobin | 9547 (56.99***)[e] | 7608 (61.50***)[b] | 9632 (64.51***)[c] | 8722 (57.95***) | -20.31 | +0.89 | -8.64 |
| Inappropriate blood pressure, heart rate, or rhythm | 1528 (9.12***)[g] | 1409 (11.39***)[b] | 1373 (9.20***)[c] | 1410 (9.37***) | -7.79 | -10.14 | -7.72 |
| Surgery operation and/or intervention | 676 (4.04***)[a] | 429 (3.47***)[b] | 786 (5.26***) | 905 (6.01***) | -36.54 | +16.27 | +33.88 |
| Medication usage | 421 (2.51***)[a] | 256 (2.07***)[c] | 300 (2.01***)[c] | 486 (3.23***) | -39.19 | -28.74 | +15.44 |
| Tattoos and/or body piercing | 797 (4.76***)[g] | 459 (3.71***)[b] | 658 (4.41***) | 699 (4.64***) | -42.41 | -17.44 | -12.30 |
| Refusal to donate | 217 (1.30***)[b] | 148 (1.20***)[f] | 106 (0.71***)[c] | 160 (1.06***) | -31.80 | -51.15 | -26.27 |
| Inappropriate body weight and/or age | 118 (0.70***)[a] | 59 (0.48***)[b] | 74 (0.50***)[c] | 53 (0.35***) | -50.0 | -37.29 | -55.08 |
| Other reasons | 3447 (20.58***) | 2003 (16.19***) | 2001 (13.40***) | 2617 (17.39***) | -41.89 | -41.95 | -24.08 |
| All | 16751 | 12371 | 14930 | 15052 | -26.15 | -10.87 | -10.14 |

*Deferral rate

**From total permanent deferrals during particular year***From total temporary deferrals during particular year

[a]–P<0.05 comparing with the 1-st, 2-nd and 3-rd pandemic years

[b]-P<0.05 comparing with the 2-nd and 3-rd pandemic years

[c]-P<0.05 comparing with the 3-rd pandemic year

[e]-P<0.05 comparing with the 1-st and 2-nd pandemic years

[f]-P<0.05 comparing with the 2-nd pandemic year

[g]-P<0.05 comparing with the 1-st pandemic year

levels of hemoglobin, inappropriate blood pressure, heart rate and/or rhythm, and recent tattoos and/or body piercing between the third pandemic and pre-pandemic years. There was a 33.88 percent increase in temporary deferrals due to recent surgery and/or intervention between the third pandemic year and the pre-pandemic year. There was a statistically significantly higher proportion of deferrals due to this reason (z = 8.1, P<0.001) during the third pandemic year compared to the pre-pandemic year. The score values of proportions' comparisons between each other year (pre-pandemic year with the first, second and third pandemic years; the first pandemic year with the second and third pandemic years; the second pandemic year with the third pandemic year, respectively) according to donors' pre-donation deferrals are provided in S3 Table.

Table 4 presents the prevalence of confirmed TTI disease markers in first-time and repeat/regular donations. There was a statistically significant (z = 3.01, P<0.05) lower prevalence of HBV infection among first-time donors during the third pandemic year than during the pre-pandemic year. Moreover, A statistically significant (z = 2.14, P<0.05) lower prevalence of all positive TTI markers among first-time donations was estimated during the third pandemic year compared with the pre-pandemic year. Finally, there was a significantly lower prevalence

**Table 4. The prevalence of confirmed transfusion-transmitted infectious disease markers in blood and its components' donations in Lithuania between April 2019 and March 2023.**

| Item | Pre-pandemic year | | 1st pandemic year | | 2nd pandemic year | | 3rd pandemic year | |
|---|---|---|---|---|---|---|---|---|
| | First-time № (%) | Repeat/ regular № (%) | First-time № (%) | Repeat/ regular № (%) | First-time № (%) | Repeat/ regular № (%) | First-time № (%) | Repeat/ regular № (%) |
| Donations | 16820 | 88501 | 9368 | 86041 | 10949 | 92682 | 11756 | 97241 |
| *Confirmed TTI markers* | | | | | | | | |
| HBV | 69 (0.410) | 7 (0.008) | 29 (0.309) | 3 (0.003) | 33 (0.301) | 6 (0.006) | 24 (0.204) | 6 (0.006) |
| HCV | 53 (0.315)[a] | 7 (0.008) | 23 (0.245) | 8 (0.009) | 24 (0.219) | 5 (0.005) | 30 (0.255) | 7 (0.007) |
| Syphilis | 31 (0.184) | 9 (0.010) | 16 (0.170) | 5 (0.006) | 22 (0.201) | 11 (0.012) | 26 (0.221) | 7 (0.007) |
| HIV 1 and 2 | 2 (0.012) | 3 (0.003) | 0 (0.0) | 5 (0.006) | 2 (0.018) | 2 (0.002) | 1 (0.009) | 5 (0.005) |
| All | 155 (0.921)[a] | 26 (0.029) | 68 (0.726) | 21 (0.024) | 81 (0.739) | 24 (0.026) | 81 (0.689) | 25 (0.025) |
| All TTI among first-time and repeat/regular donations | 181 (0.17)[b] | | 89 (0.09) | | 105 (0.10) | | 106 (0.10) | |

[a]-P<0.05 comparing with the 3-rd pandemic year

[b]-P<0.05 comparing with the 1-st, 2-nd and 3-rd pandemic year

of all positive TTI markers among all (first-time and repeat/regular) donations compared with the pre-pandemic year in the first (z = 4.8, P<0.001), second (z = 4.36, P<0.001), and third pandemic years (z = 4.72, P<0.001). The score values of prevalence comparisons of transfusion-transmitted infectious disease markers in blood and its components' donations between each other year (pre-pandemic year with the first, second and third pandemic years; the first pandemic year with the second and third pandemic years; the second pandemic year with the third pandemic year, respectively) are provided in S4 Table.

The odds ratios for confirmed TTI disease markers in all donors comparing the pre-pandemic year with the pandemic years are shown in Table 5. During the first pandemic year, the odds for HBV, HCV, and syphilis, as well as for all TTI markers, were significantly lower than those in the pre-pandemic year. Moreover, the odds for HBV and HCV and all TTI markers during the second and third pandemic years were significantly lower than those in the pre-pandemic year.

There were no statistically significant odds comparing the first pandemic year with the second and third pandemic years as well as the second pandemic year with the third pandemic year (Table 6).

**Table 5. Odds ratios for confirmed transfusion-transmitted infectious disease markers in blood and its components' donations: Comparison the pre-pandemic year with the pandemic years.**

| Confirmed infection marker | 1st pandemic year | | | 2nd pandemic year | | | 3rd pandemic year | | |
|---|---|---|---|---|---|---|---|---|---|
| | Odds ratio | 95% CIs | P-value | Odds ratio | 95% CIs | P-value | Odds ratio | 95% CIs | P-value |
| HBV | 0.47 | 0.31–0.70 | <0.001 | 0.52 | 0.35–077 | <0.001 | 0.38 | 0.25–0.58 | <0.001 |
| HCV | 0.57 | 0.37–0.88 | <0.05 | 0.49 | 0.32–0.77 | <0.05 | 0.48 | 0.31–0.75 | <0.001 |
| Syphilis | 0.57 | 0.34–0.98 | <0.05 | 0.84 | 0.53–1.33 | >0.05 | 0.80 | 0.50–1.26 | >0.05 |
| HIV 1 and 2 | 1.10 | 0.32–3.81 | >0.05 | 0.81 | 0.22–3.03 | >0.05 | 1.16 | 0.35–3.80 | >0.05 |
| All | 0.54 | 0.42–0.70 | <0.001 | 0.56 | 0.46–0.75 | <0.001 | 0.57 | 0.45–0.72 | <0.001 |

**Table 6. Odds ratios for confirmed markers of transfusion-transmitted infectious diseases in blood and its component donations in Lithuania: A comparison between pandemic years.**

| Confirmed infection marker | Comparing the first pandemic year with | | | | | | Comparing the second pandemic year with 3rd pandemic year | | |
| | 2nd pandemic year | | | 3rd pandemic year | | | | | |
| | Odds ratio | 95% CIs | P-value | Odds ratio | 95% CIs | P-value | Odds ratio | 95% CIs | P-value |
| --- | --- | --- | --- | --- | --- | --- | --- | --- | --- |
| HBV | 1.12 | 0.7–1.79 | >0.05 | 0.82 | 0.5–1.35 | >0.05 | 0.73 | 0.45–1.18 | >0.05 |
| HCV | 0.86 | 0.52–1.43 | >0.05 | 1.04 | 0.65–1.68 | >0.05 | 1.21 | 0.75–1.97 | >0.05 |
| Syphilis | 1.45 | 0.84–2.50 | >0.05 | 1.38 | 0.8–2.38 | >0.05 | 0.95 | 0.59–1.54 | >0.05 |
| HIV 1 and 2 | 0.74 | 0.2–2.74 | >0.05 | 1.05 | 0.32–3.44 | >0.05 | 1.43 | 0.4–5.05 | >0.05 |
| All TTI markers | 1.09 | 0.82–1.44 | >0.05 | 1.04 | 0.79–1.38 | >0.05 | 0.96 | 0.73–1.26 | >0.05 |

## Discussion

To the best of our knowledge, this is the first study to describe the impact of the COVID-19 pandemic on blood donation and donor profiles over an extended period, specifically covering three years of the pandemic. This survey highlights the specific challenges faced during different years of the pandemic and indicates trends toward returning to the usual pre-pandemic situation.

During the first year of the pandemic, the number of blood donations and their components decreased by 9.41 percent in Lithuania. Whole blood donations decreased by 10.88 percent compared to the pre-pandemic years. The decrease in the number of donations was lower than that in other countries [6, 17]; however, this could have been influenced by a different assessment period. When analyzing the location of blood and its component donations, it was observed that the number of donations during mobile blood drives was nearly one-third (31.8 percent) lower than that in the pre-pandemic years. Furthermore, a significant decrease in mobile blood drives was observed in other studies as well [8]. Notably, the proportion of first-time blood or component donations decreased by 44.30 percent in Lithuania during the first pandemic year, with similar trends during the second and third pandemic years. In contrast, studies in other countries showed increased first-time donations during the pandemic [7, 8].

Analysis of blood and its component donations by donor age and sex showed that the largest decrease in the number of donations was observed in the group of donors aged < 25 years. The significant decrease in donations from the youngest donor group is probably in line with the reduction in first-time donations, and the closure and transition to virtual platforms of educational facilities as well as decline of all kinds of activities in an open public places where mobile session were taking place. Similar to this study, a significant decrease was seen in donors not more than 30 years old in the USA [8]. A slight increase in the number of donations during the first pandemic year was observed only in men aged 35–44 years, 45–54 years, and 55–64 years. Generally, men were more active in donating blood and blood components during the first pandemic year, showing a small increase in proportion compared to the pre-pandemic year.

Some new actions were taken by blood establishments in Lithuania in the first months of the COVID-19 outbreak. First, blood establishments initiated a social campaign to invite and encourage persons who had recovered from COVID-19 to donate immune plasma. Blood establishments announced the testing of all potential plasma donors for SARS-CoV-2 antibodies. It could attract some donors, especially when the COVID-19 vaccination did not start yet. The interest of SARS-CoV-2 antibody status may explain the high increase of plasmapheresis as well as plasma and platelet apheresis donations during the first pandemic year in Lithuania. Nevertheless, this increase did not offset the decline in the total and first-time donations, as

evidenced by a study conducted in the USA [8]. Second, the information provided that all safety measures were in place to protect blood and its component donors from COVID-19 infection in blood establishments and collection sites during mobile sessions. The information regarding additional safety measures has been spread on regular and social media, and websites of blood establishments. The importance of this aspect demonstrated the study in Australia, revealing a high donor interest in the specific safety measures that blood services are putting in place to protect them and other donors [18]. Third, the blood establishments took an enhanced personalized approach to the donors. The personal invitations to donate blood or its components through direct communication with donors were intensified. Furthermore, for a specific duration, the blood establishments provided an individualized transportation service for donors, facilitating transport to the blood establishment and back. Reinforced attention to regular and repeat donors and diminished recruitment of new donors could explain, at least in part, why the number of first–time donors (including those under 25 years old) and donations have declined in Lithuania.

During the second year of the pandemic, donations of blood and its components in Lithuania nearly reached pre-pandemic levels, and in the third year, donations were 3.45 percent higher than in pre-pandemic years. The number of donations during mobile blood drives started to increase in the second and third years of the pandemic but remained at a lower level compared to the pre-pandemic year. When evaluating the decreasing trend in blood and its component donations during mobile blood drives over the pandemic period, the observed decrease did not seem to impact the overall number of donations.

Unexpectedly, the significant drop in donations in the group of donors under the age of 25 years during the first pandemic year remained almost unchanged during the second and third years. The number of first-time donations showed an increasing trend during the second and third pandemic years, after a significant drop during the first pandemic year, but remained at a very low level (30.11 percent less during the third pandemic year compared to the pre-pandemic year). The proportion of first-time donations during the third pandemic year was significantly lower than that during the pre-pandemic year. These facts are of great concern because it is important to continuously maintain an adequate donor base with a sufficient proportion of young donors [19]. During the first year of the pandemic, donations from repeat/regular donors decreased by 2.78 percent compared to pre-pandemic years. However, during the second and third years of the pandemic, they increased (4.70 and 9.86 percent, respectively) compared to the pre-pandemic years. This confirms that repeat/regular blood and blood component donors are crucial during pandemics and other force majeure situations [20–23]. Nevertheless, the pool of repeat and regular donors must be supplemented with first-time donors.

During the first pandemic year, the number of attempts to donate blood or its components decreased by 11.80 percent compared with the pre-pandemic year. The deferral rate also decreased (11.71 vs 14.03). During the third pandemic, the number of attempts to donate increased by 1.56 percent compared to the pre-pandemic year, while the deferral rate decreased. The increasing number and proportion of repeat and regular donations can probably explain this positive trend. Analyzing the trends of temporary and permanent deferrals, there was a decrease during the first pandemic year compared to the pre-pandemic year. However, the number of deferrals during the following pandemic years indicated a gradual approach toward pre-pandemic levels. A decreased donor deferral rate during the COVID-19 pandemic was also found in India [24].

The prevalence of all positive TTI markers among blood and its component donors significantly decreased during the pandemic years in Lithuania, and the estimated lower odds compared to the pre-pandemic year. First-time blood donors have a greater residual risk for

positive TTI markers than repeat donors [25–27]; thus, a decrease in positive TTI markers during the pandemic can be explained by the fact that there was a decrease in donations of first-time blood and its components. Another important aspect is that some donors may have hidden agendas in which they anticipate a gain from giving blood. They may donate blood in order to determine their infection status after exposure to an increased infection risk. Such test-seeking behavior decreases transfusion safety [28, 29]. Earlier study in Lithuania showed that 14.4 percent of donors donated blood due to an interest in getting blood test result [30]. Thus, a decrease in positive TTI markers during the pandemic in Lithuania could also be explained that the test-seeking donors may be less motivated to donate blood and its components under restricted regular human and institutional activities and (or) experience less exposures of an increased infection risk during pandemic.

In this study, statistical data on blood and its component donations, as well as statistical information on blood donors, were analyzed. Therefore, a limitation of this study is that the demand for blood and its components was not assessed. Other studies have revealed that a reduction in donor numbers has largely been matched by a decreased demand for transfusion [31] and a reduction in hospital activities, especially elective surgery and strict blood patient management, which are thought to reduce the demand for blood transfusion [32]. This study did not analyze why donors did not donate blood or its components less frequently during the COVID-19 pandemic. A survey conducted in seven European countries (Denmark, France, Germany, Italy, Portugal, the Netherlands, and the UK) showed that approximately half of all donors reported that they donated less than they would normally, which suggests a concerning drop in blood donations throughout the COVID-19 period [33].

## Conclusions

Most dimensions of blood and its component donations and donor characteristics have returned to pre-pandemic-year levels or even show positive trends. However, the concern is the decrease in donations from first-time and young donors under 25 years of age. Blood establishments need to review their strategies for recruiting new blood and blood component donors, with special attention paid to attracting young people.

Further studies are recommended to evaluate the motives of blood donors under 25 years of age and those who avoided donating during the COVID-19 pandemic.

## Supporting information

**S1 Table. The score values of donations proportions' comparisons according to specific dimensions in Lithuania between April 2019 and March 2023.**
(DOCX)

**S2 Table. The score values of donations proportions' comparisons according to donor's age and sex in different years in Lithuania between April 2019 and March 2023.**
(DOCX)

**S3 Table. The score values of proportions' comparisons of blood and its components donors' deferrals in Lithuania between April 2019 and March 2023.**
(DOCX)

**S4 Table. The score values of prevalence comparisons of transfusion-transmitted infectious disease markers in blood and its components' donations in Lithuania between April 2019 and March 2023.**
(DOCX)

## Author Contributions

**Conceptualization:** Vytenis Kalibatas, Lina Kalibatienė, Dulat Imashpayev.

**Data curation:** Lina Kalibatienė.

**Formal analysis:** Vytenis Kalibatas, Lina Kalibatienė, Dulat Imashpayev.

**Methodology:** Vytenis Kalibatas.

**Resources:** Lina Kalibatienė.

**Supervision:** Vytenis Kalibatas.

**Visualization:** Lina Kalibatienė.

**Writing – original draft:** Lina Kalibatienė, Dulat Imashpayev.

**Writing – review & editing:** Vytenis Kalibatas.

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
