## [Editor Report · Decision Letter 0]

26 Sep 2023

PONE-D-23-28681Blood donations and donors' profile in Lithuania: trends for coming back after COVID-19 outbreak?PLOS ONE

Dear Dr. Kalibatas,

Thank you for submitting your manuscript to PLOS ONE. After careful consideration, we feel that it has merit but does not fully meet PLOS ONE’s publication criteria as it currently stands. Therefore, we invite you to submit a revised version of the manuscript that addresses the points raised during the review process.

We look forward to receiving your revised manuscript.

Kind regards,

Enoch Aninagyei, PhD

Academic Editor

PLOS ONE

Journal Requirements:

2. Please amend either the title on the online submission form (via Edit Submission) or the title in the manuscript so that they are identical.

Additional Editor Comments:==============================1. Remove the Objective subheading on page 12 of the PDF document. Merge the content under the objective with the introduction2. Write the Materials and Methods section in themes

---

## [Author Response · Author response to Decision Letter 0]

5 Oct 2023

Editors 

PLOS ONE 

October 5, 2023

Dear Editor, 

We thank you and the reviewers for a thorough reading and constructive suggestions to improve our manuscript and for the opportunity to revise and resubmit it. We are pleased to submit the improved research article “Blood donations and donors' profile in Lithuania: trends for coming back after the COVID-19 outbreak?“ for your consideration for publication in PLOS ONE. Below you will find our response to the Editor comments. 

On behalf of the authors, I thank you for your consideration of this resubmission. We appreciate your time and look forward to your response. 

Sincerely,

Prof. Vytenis Kalibatas, MD, MPH, PhD (corresponding author)

Department of Health Management,

Lithuanian University of Health Sciences

Tilžės str. 18, LT-47181, Kaunas, Lithuania

E-mail: vytenis.kalibatas@lsmuni.lt

Journal Requirements

We thank you for the advices on the revision. We followed the templates and changed the first page and level 2 headings in “Material and Methods” section of the manuscript according to the requirements. 

2. Please amend either the title on the online submission form (via Edit Submission) or the title in the manuscript so that they are identical.

We changed the title on the online submission form. 

We reviewed the reference list, and it is complete and correct. To the best of our knowledge, we did not cite any retracted article. 

We have cited only one document, which has been canceled, i.e. Decision of the Government of the Republic of Lithuania on 14-03-2020 № 270, declaring a quarantine in the territory of the Republic of Lithuania (Reference № 13). The rationale for citing this reference is explained in the manuscript Material and Methods section: due to the particular period of declaring the quarantine, we determined that the one-year study period would start on April 1 and end on March 31. We added information that this Decision was later canceled, and provided a new reference to the document canceling this legal act.

4. Responses to Additional Editor Comments: 

4.1. Remove the Objective subheading on page 12 of the PDF document. Merge the content under the objective with the introduction 

Done. The Objective subheading is removed, and content is merged with the Introduction section.

4.2.. Write the Materials and Methods section in themes 

Done. Materials and Methods section is divided into subsections: Data collection, Study period, Donor screening and deferral, Statistical analysis and Ethical considerations.

---

## [Editor Report · Decision Letter 1]

10 Oct 2023

PONE-D-23-28681R1Blood donations and donors' profile in Lithuania: trends for coming back after the COVID-19 outbreak?PLOS ONE

Dear Dr. Kalibatas,

Thank you for submitting your manuscript to PLOS ONE. After careful consideration, we feel that it has merit but does not fully meet PLOS ONE’s publication criteria as it currently stands. Therefore, we invite you to submit a revised version of the manuscript that addresses the points raised during the review process.

ACADEMIC EDITOR:Upload a clean version of the manuscript

We look forward to receiving your revised manuscript.

Kind regards,

Enoch Aninagyei, PhD

Academic Editor

PLOS ONE
---

## [Author Response · Author response to Decision Letter 1]

11 Oct 2023

Editors 

PLOS ONE 

October 11, 2023

Dear Editor, 

We thank you and the reviewers for a thorough reading and for the opportunity to resubmit it. We are pleased to submit the improved research article “Blood donations and donors' profile in Lithuania: trends for coming back after the COVID-19 outbreak?“ for your consideration for publication in PLOS ONE. 

Response to Academic Editor:

Upload a clean version of the manuscript – The new version of the manuscript is uploaded. 

On behalf of the authors, I thank you for your consideration of this resubmission. We appreciate your time and look forward to your response. 

Sincerely,

Prof. Vytenis Kalibatas, MD, MPH, PhD (corresponding author)

Department of Health Management,

Lithuanian University of Health Sciences

Tilžės str. 18, LT-47181, Kaunas, Lithuania

E-mail: vytenis.kalibatas@lsmuni.lt

---

## [Decision Letter · Decision Letter 2]

26 Oct 2023

PONE-D-23-28681R2Blood donations and donors' profile in Lithuania: trends for coming back after the COVID-19 outbreak?PLOS ONE

Dear Dr. Kalibatas,

Thank you for submitting your manuscript to PLOS ONE. After careful consideration, we feel that it has merit but does not fully meet PLOS ONE’s publication criteria as it currently stands. Therefore, we invite you to submit a revised version of the manuscript that addresses the points raised during the review process.

We look forward to receiving your revised manuscript.

Kind regards,

Enoch Aninagyei, PhD

Academic Editor

PLOS ONE

Journal Requirements:

Reviewers' comments:

Reviewer's Responses to Questions

**Comments to the Author**

1. If the authors have adequately addressed your comments raised in a previous round of review and you feel that this manuscript is now acceptable for publication, you may indicate that here to bypass the “Comments to the Author” section, enter your conflict of interest statement in the “Confidential to Editor” section, and submit your "Accept" recommendation.

Reviewer #1: (No Response)

Reviewer #2: (No Response)

Reviewer #3: (No Response)

Reviewer #4: (No Response)

2. Is the manuscript technically sound, and do the data support the conclusions?

Reviewer #1: Partly

Reviewer #2: Yes

Reviewer #3: Yes

Reviewer #4: Partly

3. Has the statistical analysis been performed appropriately and rigorously? 

Reviewer #1: I Don't Know

Reviewer #2: Yes

Reviewer #3: Yes

Reviewer #4: N/A

4. Have the authors made all data underlying the findings in their manuscript fully available?

Reviewer #1: No

Reviewer #2: Yes

Reviewer #3: (No Response)

Reviewer #4: Yes

5. Is the manuscript presented in an intelligible fashion and written in standard English?

Reviewer #1: Yes

Reviewer #2: Yes

Reviewer #3: Yes

Reviewer #4: Yes

6. Review Comments to the Author

Reviewer #1: In this manuscript the following points deserve attention:

1)In Statistical analysis it should be written as IBM SPSS software (version 29) instead of SPSS

2)In page no 13(146,147,148,149) ,this analysis is not mentioned anywhere

3)In each 'Table' z score and p-value should be included

4)Operational definition of 'regular donor' should be clarified

5)In Table 5 pre-pandemic year Odds ratio,95% CI, p-value not mentioned

Otherwise, it is a good well-written study.

Reviewer #2: This is an important manuscript with relevant public health implications that could assist public health experts to plan during pandemic situations to ensure sustainable blood stocks. The manuscript is well written. However, the discussion lacks a much needed critical engagement of the data in the light of other studies to provide a context for the data. The authors should consider addressing issues raised below to improve the manuscript.

Materials and methods

1. Line 121 (Statistical analyses): Authors should consider adding extra bit of information to bring clarity to the assumptions for the data analyses. From the results and tables presented in the manuscript, the authors used the pre-pandemic blood donation stocks as the baseline to which the pandemic years (1, 2, 3) were compared to. Please consider making this explicit under the statistical analyses section.

2. Line 121 (statistical analyses) - Is there any reason why the pandemic years were not compared to each other? as in multiple comparisons? How does the reader judge whether the changes observed in pandemic year 1 significantly differed from pandemic year 2 or 3? This might become important when discussing as to whether specific public health policies implemented in Lithuania during the pandemic had significant change in public perception to blood donation for example.

RESULTS

3.The descriptions of Table 1 in Line 162 - 169 should precede Table 2, but not come after Table 2.

4. The statement "During the pandemic, the number of pre-donation deferrals decreased (Table 3)" in line 190 - 191 does not seem to be a complete statement.

5. The descriptions to table line 228 - 232 should come after table 4.

DISCUSSION

1. The discussion could be improved if the authors critically engage the data, particularly to highlight plausible reasons as to why there were differences in the data reported herein compared to studies from other places. For example, in line 261-262, the authors references studies that gave contrasting findings, whereas in line in line 258-259 (and line 303), the authors cite studies that gave similar results. What might have accounted for these differences in the reported studies? For example, were there differences in easing of restrictions in Lithuania compared to other countries? Even in this data being reported, what were plausible explanations for the increasing trend towards donations after the dip in the first pandemic year? Could this be attributed to public health campaigns? What led to improved public perceptions to blood donation in Lithuania? What might have the blood collection centers in Lithuania done differently during the course of the pandemic to have led to improved blood donation trend.

2. Again, why was the blood donation trends in the <25 years (in this study) very different from the other age groups?

Since no one could predict when another pandemic might strike, critical exploration of the data that provides enabling context will enable public health experts in other countries to tap into this resource to ensure sustained blood availability of blood in the event of another pandemic

Reviewer #3: The section of ethical consideration needs a minor revision.

Titles of the tables need revision and suggestions have been given.

Authors should include a "trend" graph in addition to the table 1. This will display the subject under consideration more appropriately.

Reviewer #4: Comment to the respected authors:

-As an expert in this field, I found this report very informative and in appropriate compliance with the field of transfusion medicine. However, I think the paper would be better reformatted as a short scientific report rather than a full length article to have a better chance of acceptance.

-I also have a specific recommendation to better interpret the important finding of this study regarding the observed lower prevalence of transfusion-transmitted infectious markers (TTI) cases among donations compared to the year before the pandemic.

I may somewhat be agree with the respected authors that this can be due to a decrease in the number of first-time blood donor during pandemic ,but according to my years of experience, there are always some donors who sometimes want to donate blood to monitor their health status, some of whom have been exposed to high-risk sexual behaviors. However, such people may be less motivated to donate in difficult pandemic conditions than regular donors and therefore it is very likely that the number of such donors has decreased significantly during the COVID-19 pandemic, which is reflected in the decrease in positive cases.

On the other hand, the reduction of high-risk sexual behaviors during the pandemic can also be another reason for this finding.

In my opinion, these are important issues that should be properly addressed in the discussion of this manuscript.

7. PLOS authors have the option to publish the peer review history of their article (what does this mean?). If published, this will include your full peer review and any attached files.

Reviewer #1: **Yes: **Sushanta Kumar Basak

Reviewer #2: **Yes: **Dr Patrick Adu

Reviewer #3: **Yes: **Ochaka Julie Egesie

Reviewer #4: No

---

## [Author Response · Author response to Decision Letter 2]

6 Dec 2023

Editors 

PLOS ONE 

December 4, 2023

Dear Editor, 

We thank you and the reviewers for a thorough reading and constructive comments to improve our manuscript and for the opportunity to revise and resubmit it. We are pleased to submit the improved research article “Blood donations and donors' profile in Lithuania: trends for coming back after the COVID-19 outbreak?“ for your consideration for publication in PLOS ONE. Below you will find our responses to Reviewers. 

On behalf of the authors, I thank you for your consideration of this resubmission. We appreciate your time and look forward to your response. 

Sincerely,

Prof. Vytenis Kalibatas, MD, MPH, PhD (corresponding author)

Department of Health Management,

Lithuanian University of Health Sciences

Tilžės str. 18, LT-47181, Kaunas, Lithuania

E-mail: vytenis.kalibatas@lsmuni.lt

Responses to Reviewers: 

№ Comments Responses 

Reviewer #1 

1. In Statistical analysis it should be written as IBM SPSS software (version 29) instead of SPSS 

Answer:

Accepted. We added the missed word in the sentence. Thank you.

2. In page no 13(146,147,148,149), this analysis is not mentioned anywhere 

Answer: 

Done. We added an additional information on remunerated and voluntary non-remunerated donations in the Table 1. 

3. In each 'Table' z score and p-value should be included 

Answer:

Done. We added an additional information on the presence of statistically significant differences while compared proportions between each other years in each table, and included 4 additional supplement tables with the score values. 

4. Operational definition of 'regular donor' should be clarified 

Answer: 

Done. We provided the definitions of first-time, repeat and regular donor in Donor screening and deferral section. 

5. In Table 5 pre-pandemic year Odds ratio, 95% CI, p-value not mentioned 

Answer: The odds ratio is the set of two odds. In our case – the odds of confirmed TTI in pre-pandemic year versus the odds of confirmed TTI in the first pandemic (second pandemic, third pandemic) year. For example, the odds ratio 0.47 for HBV in the first-pandemic year (Table 5) means that there is a 47% decrease in the odds of HBV during first pandemic year, compared with the pre-pandemic year. Thus, there is no point in calculating and mentioning the odds ratio for the same year (pre-pandemic versus pre-pandemic) since the odds ratio and P values will be the same, i.e., equal to 1.00.

Thank you for all remarks!

Reviewer #2

1. Line 121 (Statistical analyses): Authors should consider adding extra bit of information to bring clarity to the assumptions for the data analyses. From the results and tables presented in the manuscript, the authors used the pre-pandemic blood donation stocks as the baseline to which the pandemic years (1, 2, 3) were compared to. Please consider making this explicit under the statistical analyses section. 

Answer: 

Considering all your comments, we have opted to include statistical information for comparisons not only between the pre-pandemic year and all the pandemic years but also among the different pandemic years. Otherwise, we would have done the suggested corrections in the Statistical analysis section.

Thank you.

2. Line 121 (statistical analyses) - Is there any reason why the pandemic years were not compared to each other? as in multiple comparisons? How does the reader judge whether the changes observed in pandemic year 1 significantly differed from pandemic year 2 or 3? This might become important when discussing as to whether specific public health policies implemented in Lithuania during the pandemic had significant change in public perception to blood donation for example. 

Answer: 

Done. 

We fully acknowledge the significance of this issue and, in response to your suggestion, have included additional information on comparisons among the different pandemic years. Initially, our intent was to avoid overloading the manuscript with statistical numbers; however, your insightful comments have led us to include this important information (refer to Tables 1-4 and the new Table 6, along with supplementary details in the Results section).

We sincerely appreciate your valuable remark. Thank you very much for bringing this to our attention.

3. The descriptions of Table 1 in Line 162 - 169 should precede Table 2, but not come after Table 2. 

Answer:

Done. 

4. The statement "During the pandemic, the number of pre-donation deferrals decreased (Table 3)" in line 190 - 191 does not seem to be a complete statement. 

Answer: 

Done.

We corrected the sentence as follows: “During the pandemic, the number of pre-donation deferrals decreased comparing with the pre-pandemic year” (line 248-249). 

Thank you.

5. The descriptions to table line 228 - 232 should come after table 4.

Answer: Done. 

6. The discussion could be improved if the authors critically engage the data, particularly to highlight plausible reasons as to why there were differences in the data reported herein compared to studies from other places. For example, in line 261-262, the authors references studies that gave contrasting findings, whereas in line in line 258-259 (and line 303), the authors cite studies that gave similar results. What might have accounted for these differences in the reported studies? For example, were there differences in easing of restrictions in Lithuania compared to other countries? Even in this data being reported, what were plausible explanations for the increasing trend towards donations after the dip in the first pandemic year? Could this be attributed to public health campaigns? What led to improved public perceptions to blood donation in Lithuania? What might have the blood collection centers in Lithuania done differently during the course of the pandemic to have led to improved blood donation trend. 

Answer: 

Done. We acknowledge the importance of the issues you have raised. In response to your insightful comments, we have made significant additions to the Discussion section. Specifically, we have detailed the actions taken during the pandemic in Lithuania, which not only shed light on crucial aspects of our findings but also facilitate comparisons with actions taken in other countries.

We greatly appreciate your valuable remarks.

7. Again, why was the blood donation trends in the <25 years (in this study) very different from the other age groups? 

Answer: Regrettably, we cannot directly address this crucial question. However, we can offer some general reflections, such as the closure and transition to virtual platforms of educational facilities, along with the decline of activities in open public places where mobile sessions were previously conducted. Additionally, we posit that reinforced attention to regular and repeat donors coupled with diminished recruitment of new donors could, at least in part, explain the decline in the number of first-time donors (including those under 25 years old), and donations in Lithuania (please see the Discussion section). Therefore, we recommended further studies on evaluation of the motives of blood donors under 25 years of age and those who avoided donating during the COVID-19 pandemic (please see Conclusions).

Thank you for all remarks!

Reviewer #3

1. The section of ethical consideration needs a minor revision. 

Answer: 

Accepted. We added the proposed sentence at the beginning of Ethical consideration section. Thank you. 

2. Titles of the tables need revision and suggestions have been given. 

Answer: 

Accepted. We reformulated the titles of the tables according to the suggestions. Thank you.

3. Authors should include a "trend" graph in addition to the table 1. This will display the subject under consideration more appropriately. Answer:

Done. We included Figure 1, which shows the total number of blood and blood components’ donations, the number of first-time and repeat/regular donations during pre-pandemic and pandemic years.

Thank you!

Reviewer #4

1. I think the paper would be better reformatted as a short scientific report rather than a full length article to have a better chance of acceptance 

Answer: 

Thank you for your insightful comments. We appreciate your observation. The decision to categorize our manuscript as a full-length article was based on the consideration of numerous indicators and their values spanning a four-year period. Our intention was to present comprehensively these indicators to facilitate a thorough understanding and evaluation of the overall situation. We acknowledge the significance of your suggestion and we will incorporate it into consideration while preparing the new manuscript.

2. I also have a specific recommendation to better interpret the important finding of this study regarding the observed lower prevalence of transfusion-transmitted infectious markers (TTI) cases among donations compared to the year before the pandemic. I may somewhat be agree with the respected authors that this can be due to a decrease in the number of first-time blood donor during pandemic, but according to my years of experience, there are always some donors who sometimes want to donate blood to monitor their health status, some of whom have been exposed to high-risk sexual behaviors. However, such people may be less motivated to donate in difficult pandemic conditions than regular donors and therefore it is very likely that the number of such donors has decreased significantly during the COVID-19 pandemic, which is reflected in the decrease in positive cases. 

Answer:

Done. We acknowledge the significance of addressing the issue of test-seeking donors in the context of the decline in Transfusion-Transmissible Infections (TTI) among blood and its components donors during the pandemic. We could not find any publication on the behavior and motivation of test-seeking blood donors specifically during the COVID-19 pandemic. Nevertheless, we have incorporated relevant information in the Discussion section (lines 412-416) when deliberating on the observed decrease in TTI among donors during the pandemic.

We appreciate your astute observation and the importance of this aspect, and we thank you for bringing it to our attention.

3. On the other hand, the reduction of high-risk sexual behaviors during the pandemic can also be another reason for this finding. 

Answer:

Done. We appreciate and agree with your observation regarding the decline of TTI among blood donors during the pandemic, suggesting a possible link to a reduction in high-risk behavior. Despite an exhaustive literature search, we were unable to find supporting information for this particular assertion. Conversely, our investigation did reveal information indicating an increase in high-risk behavior among certain demographic groups during the pandemic, although not specifically among blood donors. Nevertheless, we have addressed this aspect in the Discussion section (lines 416-419).

Thank you for all remarks and suggestions!

---

## [Editor Report · Decision Letter 3]

18 Dec 2023

PONE-D-23-28681R3Blood donations and donors' profile in Lithuania: trends for coming back after the COVID-19 outbreak?PLOS ONE

Dear Dr. Kalibatas,

Thank you for submitting your manuscript to PLOS ONE. After careful consideration, we feel that it has merit but does not fully meet PLOS ONE’s publication criteria as it currently stands. Therefore, we invite you to submit a revised version of the manuscript that addresses the points raised during the review process.

**ACADEMIC EDITOR: **

Title: Considering rephrasing not in a question form

Abstract

Line 35: Indicate the full forms of all abbreviations on first mention

Lines 31/39: under 25 what? Years?

Methods

Line 89: Indicate the date the restrictions were eased

Lines 87-95: The pandemic was declared over, can you add the post-pandemic period as well

Line 108: italicise Treponema Pallidum

Line 134: Indicate ethics approval number

Results

Table 1 and subsequent tables, can you segregate data to include post-pandemic period?

We look forward to receiving your revised manuscript.

Kind regards,

Enoch Aninagyei, PhD

Academic Editor

PLOS ONE
---

## [Author Response · Author response to Decision Letter 3]

22 Dec 2023

We thank you for a thorough reading and constructive comments to improve our manuscript and for the opportunity to revise and resubmit it. We are pleased to submit the improved research article “Blood donations and donors' profile in Lithuania: trends for coming back after the COVID-19 outbreak“ for your consideration for publication in PLOS ONE. Below you will find our responses to Academic editor comments. 

On behalf of the authors, I thank you for your consideration of this resubmission. We appreciate your time and look forward to your response. 

Responses to comments

1. Title: Considering rephrasing not in a question form 

Done

2. Abstract. Line 35: Indicate the full forms of all abbreviations on first mention 

Done

3. Abstract. Lines 31/39: under 25 what? Years? 

Corrected in the Abstract and other sections of the manuscript

4. Methods. Line 89: Indicate the date the restrictions were eased 

Done

5. Methods. Lines 87-95: The pandemic was declared over, can you add the post-pandemic period as well 

We can theoretically assume that the beginning of the post-pandemic period starts after May 5, 2023. On May 5, 2023, the WHO Emergency Committee on COVID-19 recommended recognizing that, given that the disease was well-established and ongoing, it no longer fit the definition of a Public Health Emergency of International Concern (PHEIC). Unfortunately, we could not find any official statement or scientific publication on the earlier beginning of “post-pandemic” period, which could cover the period of our study. Our study covered the period from April 1, 2019 until March 31, 2023, so we did not analyze the data in the post-pandemic period (according to WHO statement on May 5, 2023). 

6. Methods. Line 108: italicise Treponema Pallidum 

Done

7.Methods. Line 134: Indicate ethics approval number 

The Kaunas Regional Biomedical Research Ethics Committee did not issue an ethics approval number, since this survey is not the subject of bioethical regulation (because generalized, fully anonymized, and publicly available data have been used). As outlined in the Ethical considerations section, “The Kaunas Regional Biomedical Research Ethics Committee reviewed the methodology of this survey and deemed the investigation an evaluation of service, not requiring review by an ethics committee, as the direct object of this study was not a specific person or person´s health". Due to misinterpretation, we removed the first sentence in the Ethical consideration section.

8. Results. Table 1 and subsequent tables, can you segregate data to include post-pandemic period? 

We acknowledge the concern regarding the post-pandemic period, and we were ready to segregate and re-calculate data in order to include the "post-pandemic" period. But, as previously outlined, we could not find any other official statement (except the WHO statement on May 5, 2023) or scientific publication on the beginning of “post-pandemic” period, which could cover the period of our study. Nevertheless, the primary focus of our study was to assess the number, types, and safety of blood and its components' donations, along with the demographic profile of blood donors. Despite how we name a year (or shorter period) after the COVID-19 outbreak, our approach involved a comparison of each pre-pandemic year with subsequent years of post-COVID-19 outbreak, ensuring statistically reliable comparisons across equivalent time periods.

Thank you very much for all remarks and comments!

Sincerely,

Prof. Vytenis Kalibatas, MD, MPH, PhD (corresponding author)

Department of Health Management,

Lithuanian University of Health Sciences

Tilžės str. 18, LT-47181, Kaunas, Lithuania

E-mail: vytenis.kalibatas@lsmuni.lt

---

## [Editor Report · Decision Letter 4]

9 Jan 2024

Blood donations and donors' profile in Lithuania: trends for coming back after the COVID-19 outbreak

PONE-D-23-28681R4

Dear Dr. Vytenis Kalibatas,

We’re pleased to inform you that your manuscript has been judged scientifically suitable for publication and will be formally accepted for publication once it meets all outstanding technical requirements.

Kind regards,

Enoch Aninagyei, PhD

Academic Editor

PLOS ONE
---

## [Editor Report · Acceptance letter]

16 Jan 2024

PONE-D-23-28681R4 

PLOS ONE

Dear Dr. Kalibatas, 

I'm pleased to inform you that your manuscript has been deemed suitable for publication in PLOS ONE. Congratulations! Your manuscript is now being handed over to our production team.

Kind regards, 

on behalf of

Dr Enoch Aninagyei 

Academic Editor

PLOS ONE